# (−)-Epigallocatechin-3-Gallate Reduces Perfluorodecanoic Acid-Exacerbated Adiposity and Hepatic Lipid Accumulation in High-Fat Diet-Fed Male C57BL/6J Mice

**DOI:** 10.3390/molecules28237832

**Published:** 2023-11-28

**Authors:** Hong Xu, Xu Zhong, Taotao Wang, Shanshan Wu, Huanan Guan, Dongxu Wang

**Affiliations:** 1School of Grain Science and Technology, Jiangsu University of Science and Technology, Zhenjiang 212100, China; hongxuip@163.com (H.X.); zhongxu0223@163.com (X.Z.); 2Department of Clinical Nutrition, Affiliated Hospital of Jiangsu University, Zhenjiang 212000, China; jdfywtt@163.com; 3College of Agriculture & Biotechnology, Zhejiang University, Hangzhou 310058, China; wss923@163.com

**Keywords:** perfluorodecanoic acid, EGCG, endocrine-disrupting chemicals, NLRP3, SREBP−1, hepatic lipid accumulation

## Abstract

Perfluorodecanoic acid (PFDA), an enduring and harmful organic pollutant, is widely employed in diverse food-related sectors. Our previous studies have provided evidence that PFDA has the potential to facilitate obesity and hepatic fat accumulation induced by high-fat diet (HFD) intake. Epigallocatechin-3-gallate (EGCG), a polyphenol found in green tea, has been suggested to possess potential preventive effects against metabolic abnormalities and fatty liver. The purpose of this research was to investigate the effects of EGCG on PFDA-exacerbated adiposity and hepatic lipid accumulation in HFD-fed mice. The results showed that EGCG reduced body weight gain; tissue and organ weights; blood glucose, serum insulin, HOMA-IR, leptin, and lipid parameters; serum inflammatory cytokines (IL−1β, IL−18, IL−6, and TNF−α); and hepatic lipid accumulation in PFDA-exposed mice fed an HFD. Further work showed that EGCG improved liver function and glucose homeostasis in mice fed an HFD and co-exposed to PFDA. The elevated hepatic mRNA levels of SREBP-1 and associated lipogenic genes, NLRP3, and caspase−1 in PFDA-exposed mice fed an HFD were significantly decreased by EGCG. Our work provides evidence for the potential anti-obesity effect of EGCG on co-exposure to HFD and PFDA and may call for further research on the bioactivity of EGCG to attenuate the endocrine disruption effects of long-term exposure to pollutants.

## 1. Introduction

Perfluorodecanoic acid (PFDA), a noxious and enduring organic pollutant, is extensively utilized in various food-related domains, including food packaging and cookware [1,2]. Due to its widespread use, this chemical is frequently detected in many animals and in human populations worldwide, and concerns have been raised about its impact on the environment, food safety, and human health [3,4]. As PFDA is present in air, food, and water, PFDA concentrations as high as 3.21 ng/mL have been recorded in human blood samples obtained from certain Chinese cities [5]. The oral absorption of PFDA in mammals is well documented, and it accumulates primarily in the liver, with a mean serum half-life exceeding two years [6]. PFDA presents incremental toxicity due to biomagnification [6]. Exposure of rodents to PFDA resulted in liver injury and disorders in lipid metabolism, thyroid function and immune function [6,7,8,9]. Direct skin contact with consumer products and medical supplies is considered as a common route of exposure to PFDA [9]. Health concerns have led to putting restriction on its intentional addition to cosmetics in 2020.

As indicated by epidemiological studies [10,11,12], when exposed to PFDA, humans may be likely to become obese and diabetic. Dietary fat also modulates the metabolism and toxicity of perfluorinated compounds in addition to their direct health hazards effects [13]. There are numerous metabolic diseases associated with a high-fat diet (HFD), such as obesity, diabetes mellitus, coronary heart disease, fatty liver disease, and stroke [14,15,16]. These metabolic diseases may be caused by an HFD, which contributes to the production of free radicals and the development of inflammation and insulin resistance [17,18].

The nucleotide-binding domain leucine-rich repeat and pyrin domain-containing receptor 3 (NLRP3) inflammasome is a multimeric cytosolic protein complex that assembles in response to cellular perturbations [19]. As a result of this assembly, caspase−1 is activated, which promotes the maturation and release of the inflammatory cytokines interleukin−1β (IL−1β) and IL−18 [20]. Aberrant activation of the NLRP3 inflammasome can facilitate a chronic systemic low-grade inflammatory state that modulates inflammation-associated metabolic disorders [20]. The NLRP3 inflammasome engages in interactions with multiple essential functional proteins, thereby exerting influence over various cellular functions beyond the maturation of cytokines. This implies that caspase−1 possesses a broader scope of functionality beyond its role in cytokine maturation [21,22]. Patients with non-alcoholic fatty liver disease (NAFLD) exhibited elevated levels of the NLRP3 inflammasome and caspase−1, an effector protein, as a consequence of an HFD [23].

There exists a discernible correlation between NLRP3 inflammasome activation and the accumulation of lipids, as well as an elevation in the expression of the sterol regulatory element-binding protein (SREBP) 1 [24]. NLRP3 inflammasome activation plays an essential role in the development of metabolic diseases, including obesity and NAFLD, and can serve as a therapeutic target [25]. In our previous studies, we found that, as a NLRP3 inflammasome activator, PFDA promoted cellular triglyceride (TG) accumulation by upregulating SREBP 1 expression via activating the NLRP3 inflammasome in HepG2 and 3T3-L1 cell lines and HFD-fed male C57BL/6J mice [26,27]. This evidence suggest that the effects of PFDA occur as a result of the exaggeration of adipogenesis.

(−)-Epigallocatechin-3-gallate (EGCG), the most abundant catechin of green tea, accounting for 50–80% of all catechins, has been found to have strong protective effects against obesity and NAFLD [28,29,30,31,32]. Our previous study found that EGCG protected against PFDA-induced liver damage and inflammation in mice by inhibiting NLRP3 inflammasome activation [26]. In the present study, we hypothesize that PFDA exposure is involved in the promotional effects of HFD-induced obesity and hepatic lipid accumulation. As EGCG is toxicologically safe in moderate amounts, it use in the mitigation of adiposity and hepatic lipid accumulation has become popular. The present study utilized a male C57BL/6J mouse model to determine if EGCG alleviated the exaggerated consequences of PFDA exposure on obesity and lipid accumulation in the liver following HFD feeding.

## 2. Results

### 2.1. EGCG Reduced Body Weight Gain in PFDA-Exposed Mice Fed an HFD

Mice that were fed a low-fat diet (LFD) displayed typical fluctuations in body weight throughout the duration of the experiment; in addition, no significant changes in body weight were noted between the LFD-fed group and the LFD plus PFDA group (Figure 1A,B). However, mice fed an HFD demonstrated a substantial elevation in body weight starting from week 6, in contrast to the body weight of the LFD group (Figure 1A). Moreover, a significant increase in body weight was noted between the HFD-fed group and HFD-fed plus PFDA exposure group (Figure 1A). Moreover, the extent of weight gain exhibited a significant decrease in the treatment groups receiving EGCG compared to the group exposed to an HFD and PFDA (Figure 1A,B). The data depicting the food and water intake (Figure 1C,D) reveal no statistically significant variations in the consumption of food or water across the different groups. These results suggest that EGCG reduces the weight gain observed in PFDA-exposed mice fed an HFD.

### 2.2. EGCG Decreased Tissue and Organ Weights in PFDA-Exposed Mice Fed an HFD

Table 1 displays the proportions, expressed as a percentage of mouse body weight, of various organs and adipose tissues. Based on the obtained results, it is evident that there is no difference in the organ weight of the heart, spleen, and pancreas among the treatment groups. Furthermore, when compared to the HFD-fed group, the HFD-fed plus PFDA-exposed group exhibited significant increases in the weights of liver (*p* < 0.05) and kidney (*p* < 0.05) tissues. The administration of H-EGCG in response to HFD feeding plus PFDA exposure resulted in a significant decrease in liver weight (*p* < 0.05), while kidney weight remained unaffected. On the other hand, the HFD-fed group exhibited significant increases in the weights of the four types of adipose tissue (epididymal, subcutaneous, retroperitoneal, and mesenteric adipose tissue) and total adipose tissue (*p* all < 0.05) compared to the LFD-fed group. In accordance with our previous study [26], mice subjected to simultaneous exposure to an HFD and PFDA exhibited significant augmentation in the four types of adipose tissues mentioned above and total adipose tissue weights (*p* all < 0.05) as compared to mice fed an HFD alone. Furthermore, it is noteworthy that the administration of L-EGCG resulted in significant decreases in epididymal adipose tissue (*p* < 0.05) and total adipose tissue (*p* < 0.05) weights, whereas the H-EGCG led to a significant reduction in the weights of all aforementioned adipose tissue (*p* all < 0.05) in PFDA-exposed mice following HFD feeding.

### 2.3. EGCG Reduced Blood Glucose and Blood Lipid Parameters in PFDA-Exposed Mice Fed an HFD

Serum parameters were analyzed. Data are presented in Figure 2. Based on the observations depicted in Figure 2A, it is evident that the feeding of an HFD induced a notable elevation in blood glucose levels in comparison to the LFD-fed group; however, it is worth noting that EGCG treatment did not result in a significant reduction in this increase. The group of mice exposed to PFDA while being fed an HFD exhibited notably elevated levels of insulin and leptin, as well as a higher HOMA-IR index, in comparison to the group solely fed an HFD (Figure 2B–D). Moreover, the group treated with H-EGCG demonstrated noteworthy reductions in insulin and leptin serum levels, as well as HOMA-IR, compared to the group exposed to an HFD and PFDA (Figure 2B–D). These results indicate that EGCG reduces PFDA-exacerbated insulin resistance in mice fed an HFD. Based on the observations depicted in Figure 2E–G, it is evident that the increases in serum TG, TC and NEFA levels caused by the consumption of an HFD are further exacerbated by exposure to PFDA. Moreover, the group treated with H-EGCG demonstrated a noteworthy reduction in the serum concentration of NEFA, as opposed to TG and TC, compared to the group exposed to an HFD and PFDA (Figure 2E–G). Furthermore, it is apparent that there is a lack of disparity in the serum concentrations of LDL-C and HDL-C across the various treatment cohorts (Figure 2H,I).

### 2.4. EGCG Improved Liver Function in PFDA-Exposed Mice Fed an HFD

Serum ALT, AST and AKP levels were used for the evaluation of liver function, as shown in Figure 3. In contrast to the HFD-fed group, PFDA exposure resulted in significant elevations in ALT and AKP serum levels, thereby suggesting that PFDA exposure exacerbates hepatic function impairment induced by the consumption of an HFD. Furthermore, the H-EGCG treatment mice exhibited significant decreases in the serum levels of ALT and AKP compared with mice co-exposed to an HFD and PFDA (Figure 3), thereby suggesting that EGCG can mitigate the hepatic function impairment in PFDA-exposed mice fed an HFD.

### 2.5. EGCG Inhibited Serum Inflammatory Cytokines

Based on the data presented in Figure 3D, it is evident that PFDA exposure to mice elicits a significant elevation in IL−1β serum levels among both LFD-fed and HFD-fed groups. However, the administration of H-EGCG effectively mitigates the PFDA-induced increase in serum IL−1β levels in the PFDA-exposed mice fed an HFD. Similar results are also shown for IL−18. It can be seen that PFDA exposure can cause a significant increase in serum IL−18 levels in the HFD-fed mice, and L-EGCG and H-EGCG effectively inhibit the PFDA-induced increases in serum IL−18 levels in the PFDA-exposed mice fed an HFD (Figure 3E). As shown in Figure 3F, it is evident that there is no difference in serum IL−6 levels among the treatment groups. The data presented in Figure 3G clearly demonstrate that PFDA exposure leads to a substantial increase in serum TNF−α levels in HFD-fed mice. Nevertheless, the administration of L-EGCG and H-EGCG effectively attenuates the PFDA-induced elevation in TNF−α serum levels in the mice co-exposed to an HFD and PFDA.

### 2.6. EGCG Reduced Lipid Accumulation in Mice Co-Exposed to an HFD and PFDA

Histological analysis demonstrated a significant elevation in the risk of hepatic steatosis due to HFD consumption compared to LFD consumption. Furthermore, the presence of PFDA exhibited a substantial elevation in the likelihood of hepatic steatosis among mice fed an HFD, while no such effect was observed in mice fed a LFD (Figure 4A). Nevertheless, the administration of L-EGCG and H-EGCG effectively attenuates PFDA-triggered hepatic steatosis in HFD-fed mice that were exposed to PFDA (Figure 4A,C). Similarly, the utilization of oil red O staining demonstrated a stronger correlation between the intake of an HFD and elevated levels of hepatic lipid accumulation compared to the LFD group (Figure 4B). Furthermore, mice co-exposed to an HFD and PFDA exhibited an intensified level of hepatic lipid accumulation, suggesting a synergistic reaction between PFDA exposure and an HFD intake (Figure 4B). Furthermore, the H-EGCG treatment group exhibited significant decrease in lipid content in the liver compared with the HFD-fed plus PFDA-exposed group (Figure 4B,D), implying that EGCG may have the potential to mitigate hepatic lipid accumulation in mice fed an HFD and co-exposed to PFDA.

### 2.7. Effect of EGCG on Glucose Homeostasis

The impact of EGCG on glucose regulation in the mice co-exposed to an HFD and PFDA was assessed through the measurement of serum glucose levels using the insulin tolerance test (ITT) and glucose tolerance test (GTT) methods. Prior to treatment, no notable disparities were observed in the GTT and ITT outcomes (Figure 5A,D). Nevertheless, the HFD-fed mice supplemented with PFDA displayed a significantly elevated AUC in GTT compared to the HFD-fed group at week 9 (Figure 5C). Furthermore, the H-EGCG treatment group exhibited significant decrease in AUC than did the mice co-exposed to an HFD and PFDA (Figure 5C), thereby suggesting that EGCG can improve glucose intolerance in HFD-fed mice that were exposed to PFDA. During the eleventh week, the group subjected to an HFD along with exposure to PFDA exhibited notably elevated AUC values in ITT analysis, in comparison to the HFD-fed group (Figure 5F). Nevertheless, the administration of H-EGCG effectively decreases the PFDA-induced elevation in AUC in the ITT in HFD-fed mice (Figure 5F).

### 2.8. EGCG Regulated Hepatic Lipid Metabolism in PFDA-Exposed Mice Fed an HFD

We conducted an investigation to explore the mechanisms by which EGGC reduces hepatic lipid accumulation exacerbated by PFDA in mice fed an HFD, considering the significant involvement of the liver in glucose and fatty acid metabolism. As shown in Figure 6A, *SREBP-1*, *SREBP-2*, *FASN*, *SCD1*, *HMGCS2*, and *CPT-1a* mRNA expression were significantly increased in livers of the HFD + PFDA group compared with the livers of the HFD group. Furthermore, the exposure of HFD-fed mice to PFDA resulted in a significant decrease in the mRNA expression levels of *ATGL*, which plays a crucial role in fatty acid catabolism (Figure 6A). However, the examination of hepatic mRNA expression demonstrated that H-EGCG significantly attenuated the upregulation of *SREBP-1* and its associated lipogenic genes, while also enhancing the downregulation of ATGL in the HFD + PFDA group (Figure 6A). Collectively, these findings indicate that EGCG effectively inhibits the expression of the SREBP-1 pathway in mice fed an HFD and exposed to PFDA.

### 2.9. EGCG Inhibited Hepatic NLRP3 Inflammasome Expression in PFDA-Exposed Mice Fed an HFD

Given the significant involvement of NLRP3 inflammasome activation in the regulation of SREBP-1 expression, our study aimed to examine the impact of the NLRP3 inflammasome on EGGC in mitigating the exacerbated hepatic lipid accumulation induced by PFDA exposure in mice fed an HFD. As shown in Figure 6B, compared to the HFD-fed group, PFDA exposure increased the relative mRNA expression level of *NLRP3* from 1.7 fold to 3.0 fold in mice. Similarly, *caspase−1* levels increased from 53.7 fold to 80.0 fold, *IL−1β* increased from 7.4 fold to 13.9 fold, and *IL−18* increased from 6.5 fold to 15.7 fold. Following intervention with EGCG in mice fed an HFD and exposed to PFDA, the expression of all four genes (*NLRP3*, *caspase−1*, *IL−1β*, and *IL−18*) exhibited notable decreases, with the H-EGCG group demonstrating a more pronounced reduction in comparison to the L-EGCG group.

## 3. Discussion

The health risk associated with exposure to PFDA has garnered heightened attention in the past decade, primarily due to its persistent nature in the environment and its detection in substantial quantities in both animal and human blood samples [26,27]. PFDA, which bears structural resemblance to fatty acids, has been demonstrated to elicit peroxisome proliferation, disturb the equilibrium of fatty acid and cholesterol metabolism, and ultimately result in the development of a fatty liver [7]. In our previous study, PFDA potentiated HFD-induced obesity and hepatic lipid accumulation as indicated by significant hepatic lipid accumulation as well as increased body weight, inflammation, and glucose and lipid metabolism disturbances [26]. As a natural ingredient, the green tea polyphenol EGCG has been shown to attenuate obesity, fatty liver disease, and hepatic inflammation and regulate lipid profiles [32,33]. In this study, we aimed to examine the potential inhibitory effect of EGCG in PFDA-exacerbated adiposity and hepatic lipid accumulation. Our work indicates that EGCG treatment could lower PFDA-induced body weight gain, liver weight gain, and hepatic fat accumulation in mice fed an HFD. In parallel, the administration of EGCG improved the serum lipid profile, liver function, inflammation, glucose homeostasis, and hepatic glucose and fatty acid metabolism.

In this study, PFDA increased the body weight of HFD-fed mice, which is consistent with our previous studies [26]. According to the data presented in Table 1, it is evident that the increase in animal body weight primarily stems from the amplification in liver and adipose tissue weights. This observation strongly suggests that PFDA effectively promotes the additional accumulation of fat in mice fed an HFD. The weight reduction and hypolipidemic effects of EGCG have been extensively documented in the literature [31,34,35,36]. A notable study by Yuan et al. demonstrated that EGCG effectively improves lipid metabolism, mitigates inflammation, and significantly decreases FASN protein levels in the livers of rats fed an HFD [37]. Simultaneously, the serum levels of ALT and AKP serve as reliable indicators of hepatic injury and function, exhibiting a strong correlation with the extent of steatosis. The administration of EGCG resulted in reductions in ALT and AKP levels. The outcomes of oil red staining demonstrated that EGCG administration ameliorated the formation of lipid droplets. The results of our study demonstrate that the administration of EGCG effectively mitigated the metabolic profile and inhibited the accumulation of hepatic fat in mice fed an HFD and exposed to PFDA.

SREBPs are a group of transcription factors that govern lipid homeostasis through the regulation of enzyme expression necessary for the synthesis of endogenous cholesterol, fatty acids, triacylglycerol, and phospholipids [38]. The distinct functions of the three SREBP isoforms, namely SREBP-1a, SREBP-1c, and SREBP-2, in lipid synthesis have been elucidated. Experimental investigations employing transgenic and knockout mice have provided evidence that SREBP-1 plays a crucial role in fatty acid synthesis and insulin-mediated glucose metabolism, particularly in the process of lipogenesis. Conversely, SREBP-2 exhibits relatively specific involvement in cholesterol synthesis [38]. In vivo and in vitro experiments revealed that lipid accumulation was associated with significantly unregulated expression levels of the genes encoding *SREBP-1*, *FASN*, and *SCD-1* [39,40,41]. In the present study, the hepatic mRNA levels of *SREBP-1*, *SREBP-2*, and their respective lipogenic genes exhibited statistically significant increases in the group subjected to co-exposure to an HFD and PFDA, as compared to the group fed an HFD alone. These results are consistent with hepatic lipid accumulation and serum TG levels, indicating that SREBPs are involved in the metabolic process of hepatic fat accumulation in mice fed an HFD and exposed to PFDA. EGCG has been reported to regulate SREBP expression and subsequently affect liver fat metabolism in HFD-induced obese mice or rats [42,43]. In this study, we demonstrate for the first time that long-term EGCG intervention reduces PFDA-exacerbated obesity and hepatic lipid accumulation in HFD-fed mice, which is consistent with the findings in apolipoprotein E knockout mice and HFD- and streptozotocin-induced type 2 diabetic mice [44,45].

The liver is widely recognized as a primary organ affected by PFDA in both humans and rodents. Numerous studies have provided evidence of the detrimental impact of PFDA on liver enzymes and function [46,47]. In 2017, Zhou et al. reported for the first time that PFDA stimulates NLRP3 inflammasome assembly in human cells and mice tissues [48]. Subsequently, our study found for the first time that PFDA incudes liver injury and inflammation by activating that NLRP3 inflammasome and reported that EGCG protects against PFDA-induced liver injury by inhibiting the NLRP3 inflammasome in mice [26]. To date, PFDA is known to activate the NLRP3 inflammasome. In recent years, EGCG has been found to inhibit NLRP3 inflammasome-related inflammatory responses and organ damage in various animal models [49,50,51,52], and these findings revealed that EGCG is an effective natural ingredient for targeting NLRP3 inflammasome activation.

The NLRP3 inflammasome is considered a sensor of cellular homeostasis that regulates lipid metabolism, and NLRP3 inflammasome activation is associated with adipocyte differentiation and adipogenesis [53,54]. Moreover, NLRP3 inflammasome activation and NLRP3-mediated caspase−1 activation enhances adipogenic differentiation and controls adipocyte differentiation and insulin sensitivity in mesenchymal stem cells [55,56]. The present study demonstrates that HFD-fed mice exposed to PFDA exhibited significant upregulation of *NLRP3* and *caspase−1* mRNA levels along with the activation of SREBPs in the liver, which was consistent with our previous findings in HFD-fed mice that were treated with PFDA [26]. Furthermore, we demonstrated that EGCG reduces *NLRP3* and *caspase−1* mRNA levels in the livers of HFD-fed mice exposed to PFDA. Moreover, these data were consistent with our previous study on PFDA-induced liver damage and inflammation in mice [49]. Another study indicates that *Casp1*^−/−^ mice or obese wild type mice treated with a caspase−1 inhibitor exhibit increased sensitivity to insulin [55]. In this study, EGCG significantly improved glucose intolerance in HFD-fed mice exposed to PFDA, suggesting the important role of NLRP3 inhibition in the amelioration of insulin resistance in mice co-exposed to an HFD and PFDA.

A previous study investigated the impact of PFDA exposure on alterations in body weight and resting metabolic rate in a diet-induced weight-loss context, employing a 2-year randomized clinical trial design [57]. This study found a positive correlation between elevated initial plasma PFDA levels and increased weight regain, particularly among females, and suggested that PFDA may affect the prevalence of obesity. In a subsequent study conducted by this research team, it was further discovered that plasma concentrations of PFDA exhibit predominant correlations with blood lipids and apolipoproteins [58]. For the healthy adult population, it is advisable to consume dietary fat within the range of 20% to 35% of their total caloric intake [59]. Nevertheless, our findings suggest that PFDA may contribute to the accumulation of liver lipids in mice fed a diet consisting of 20% high-fat content. Therefore, we hypothesized that PFDA might increase the risks of NAFLD, especially in populations consuming HFDs. Tea is one of the most popular beverages in the world and has a huge consumer base. As one of the most abundant functional ingredients in tea, EGCG is widely present in various types of tea. The administration of 856.8 mg EGCG to women exhibiting central obesity over a period of 12 weeks yielded noteworthy outcomes, including substantial weight reduction, diminished waist circumference, and a decline in plasma total cholesterol (TC) and low-density lipoprotein (LDL) levels, all without the occurrence of any detrimental effects [60]. Similar results were also found in another clinical trial; overweight and obese women treated with green tea extracts containing 856.8 mg EGCG for 6 weeks showed effective increases in serum leptin levels and reductions in serum LDL levels [61]. Drawing from the findings of these clinical trials, it is plausible to hypothesize that PFDA could potentially serve as a significant contributing factor to the onset of obesity in individuals who regularly consume high-fat diets. Nevertheless, the consumption of green tea and EGCG may potentially play a role in mitigating the risk of obesity associated with PFDA in individuals who consume high-fat foods; however, additional clinical investigations are required to validate this assertion. The present study has several limitations that need to be addressed in additional studies. (1) Whether EGCG has the potential to mitigate obesity induced by HFD and PFDA in mice warrants further inquiry in subsequent studies. (2) Additional research is warranted to investigate the comparative impact of tea catechins and EGCG at equal doses as anti-obesity interventions.

## 4. Materials and Methods

### 4.1. Animals, Diet, and Treatment

Thirty-six 4-week-old C57BL/6J (male 18–23 g) mice were obtained from the Center of Comparative Medicine of Yangzhou University (Yangzhou, China) and maintained under constant conditions (temperature, 22 ± 3 °C; humidity, 40–50%; light/dark cycle 12 h/12 h). After one week of feeding a LFD containing 4% fat by body weight, glucose tolerance and insulin sensitivity were measured at baseline. In the next step, mice were assigned to six groups (*n* = 6/group): (1) LFD group, (2) HFD group; (3) LFD plus PFDA group; (4) HFD plus PFDA group; (5) HFD plus PFDA and low dose EGCG group; and (6) HFD plus PFDA and high-dose EGCG group. The semipurifed low-fat diet (4% fat) and high-fat diet (20% fat) were procured from Nantong Trophic Technology Co., Ltd. (Nantong, China), and the composition of the diets is provided in Table 2. Mice had access to 0.5 mg/L PFDA (98% purity, Sigma-Aldrich, St. Louis, MO, USA) aqueous solution or water for 12 consecutive weeks. During the 12-week duration, the mice in the EGCG (95% purity, Sigma-Aldrich, St. Louis, MO, USA) treatment groups were orally administered L-EGCG at a dose of 50 mg/kg and H-EGCG at a dose of 100 mg/kg once daily. Body weights were recorded weekly.

Food intake, and water intake were monitored on a weekly basis in each of the groups. Prior to the conclusion of the experiment, a 4 h fasting period was imposed on all mice. Subsequently, the mice were administered anesthesia and subsequently euthanized via cervical dislocation. Following euthanasia, the pertinent organs and adipose tissues were meticulously weighed, and their measurements were duly recorded. Plasma was collected by removing the eyeball, and serum was obtained through centrifugation of blood samples at 3000 rpm for 10 min and stored at −80 °C until analysis. A total of thirty-six liver samples were subjected to fixation in 10% (*v*/*v*) neutral-buffered formalin for the purpose of hepatic fat analysis. The aforementioned samples were subsequently subjected to paraffin sectioning. Conversely, the remaining liver samples were preserved at a temperature of −80 °C until they could be analyzed using freezing techniques facilitated by liquid nitrogen. All experiments were approved by the Institutional Animal Care and Use Committee of Jiangsu University of Science and Technology (protocol code: 20200302).

### 4.2. Quantification of Serum Parameters

Serum levels of TG, total cholesterol (TC), glucose, nonesterified fatty acid (NEFA), low-density lipoprotein cholesterol (LDL-C), and high-density lipoprotein cholesterol (HLD-C) were measured using the purchased commercial kits from Nanjing Jiancheng Bioengineering Institute (Nanjing, China) in accordance with the manufacturer’s instructions. The levels of insulin in mouse serum were quantified using enzyme-linked immunosorbent assay (ELISA) kits obtained from Thermo Fisher (Waltham, MA, USA). The quantification of leptin levels in mouse serum was performed using ELISA kits sourced from R&D (Minneapolis, MH, USA).

### 4.3. Liver Function Analysis

Serum levels of alanine aminotransferase (ALT), aspartate aminotransferase (AST), and alkaline phosphatase (AKP) were measured using commercial kits purchased from Nanjing Jiancheng Bioengineering Institute (Nanjing, China) in accordance with the manufacturer’s instructions.

### 4.4. Serum Inflammatory Cytokine Quantification

Interleukin (IL)−1β and IL−6 contents in serum were measured using commercial ELISA kits purchased from Sigma-Aldrich (St. Louis, MO, USA) in accordance with the manufacturer’s instructions. The quantification of IL−18 and tumor necrosis factor (TNF)-α levels in mouse serum was performed using ELISA kits sourced from Invitrogen (Carlsbad, CA, USA), following the guidelines provided by the manufacturer.

### 4.5. Metabolic Testing

A 4 h fast, insulin tolerance test (ITT) was conducted on experimental mice during the adaptation period and at weeks 5 and 9 following treatments. Blood glucose levels were measured using a blood glucose meter at 0, 15, 30, 60, and 120 min after the intraperitoneal injection of a human recombinant insulin (Sigma-Aldrich, St. Louis, MO, USA) solution (0.75 U/kg body weight). Additionally, a 6 h fast, intraperitoneal glucose tolerance test (GTT) was performed on all mice during the adaptation period and at weeks 6 and 11 after treatments. A 20 percent intraperitoneal D-(+)-glucose (Sigma-Aldrich, St. Louis, MO, USA) solution (2 g/kg body weight) was administered, and blood glucose levels were assessed concurrently with the ITT post-administration. A homeostasis model assessment-insulin resistance (HOMA-IR) calculator [HOMA-IR = [fasting insulin (mU/L) × fasting blood glucose (mmol/L)]/22.5] was used to determine the HOMA-IR score. The areas under the curve (AUCs) for ITT and GTT were determined using GraphPad Prism 5.0 (San Diego, CA, USA).

### 4.6. The Quantification of Lipid Accumulation within the Hepatic Tissue

The hepatic lipid contents and hepatic adipocyte morphology were assessed with Oil Red O staining and hematoxylin and eosin (H&E) staining as previously described [26]. Lipid droplets in the samples were quantified using ImageJ software (V1.53c, NIH, Bethesda, MD, USA).

### 4.7. mRNA Expression Analysis

Total RNA was isolated from liver tissues using a TRIzol reagent (Takara Biotechnology, Dalian, China) according to the manufacturer’s protocol. Total RNA was quantified using a NanoDrop Microvolume Spectrophotometer (Thermo Fisher Scientific Inc., Waltham, MA, USA). cDNA was obtained using total RNA, oligo dT primer, and the PrimeScript RT Enzyme Mix following the manufacturer’s instructions (Takara Biotechnology, Dalian, China). Relative gene expressions were quantified using RT-qPCR analysis (Real Time PCR CFX System; Bio-Rad, Hercules, CA, USA) with a SYBR Green kit. The sequences are shown in Table 3, and primers were synthesized by Generay Biotechnology (Shanghai, China). Based on the manufacturer’s specifications, the relative gene expression was evaluated by calculating 2^−ΔΔCq^.

### 4.8. Statistical Analysis

The results are presented as mean ± standard errors of the mean (SEM). One-way analysis of variance (ANOVA) followed by Tukey’s test was used for comparisons between multiple groups. For the results presented in Figure 1A, two-way ANOVAs were used to determine the differences in each group. Values were considered significantly different when *p* < 0.05.

## 5. Conclusions

Taken together, this study provides evidence that long-term PFDA exposure significantly increased body weight, increased hepatic lipid accumulation, impaired glucose homeostasis, and increased mRNA expression of SREBPs and their target genes as well as the NLRP3 inflammasome, thereby contributing partly to the exacerbation of steatohepatitis in HFD-induced mice at the doses indicated. Moreover, this study revealed the significant protective effects of EGCG against PFDA-exacerbated obesity and hepatic lipid accumulation in HFD-fed mice. As for the mechanism, EGCG possessed considerable lipid-lowering activity that enabled it to downregulate SREBP pathway expression by inhibiting NLRP3 inflammasome activation (Figure 7). Our findings may promote further understanding of the mechanism underlying the bioactivity of EGCG to attenuate the endocrine disruption effects of long-term exposure to chemicals and draw attention to its promising potential for dietary application. Although further research is required to elucidate the efficacy and safety of long-term EGCG use in humans, caution is needed when administering EGCG supplements for weight management.

## Figures and Tables

**Figure 1 molecules-28-07832-f001:**
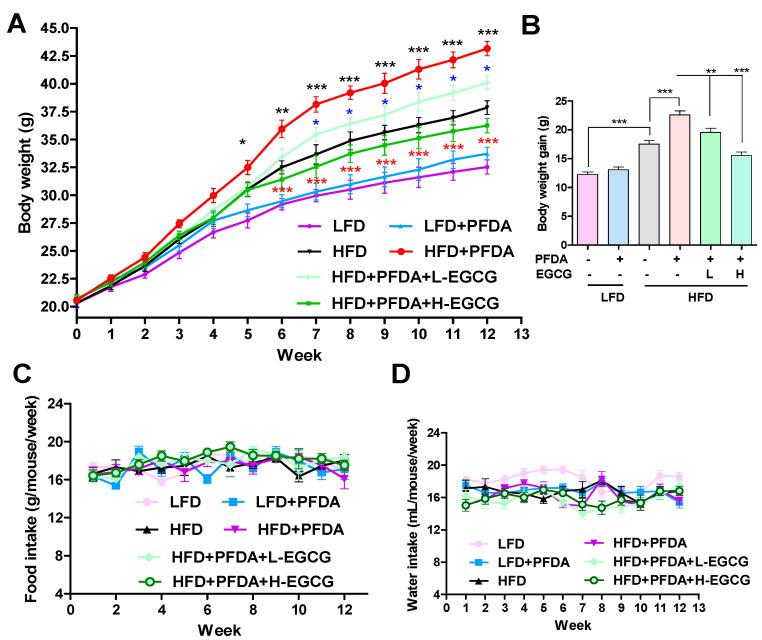
Effects of EGCG on body weight, body weight gain, and food and water intake in PFDA-exposed mice fed an HFD. (**A**) Body weight monitored weekly in each group; (**B**) Body weight gain for 12 weeks in each group; (**C**) Food intake monitored weekly in each group; (**D**) Water intake monitored weekly in each group. Numbers are the mean ± SEM (*n* = 6/group). HFD vs. HFD + PFDA: * *p* < 0.05, ** *p* < 0.01 or *** *p* < 0.001 indicated by black asterisk; HFD + PFDA vs. HFD + PFDA + L-EGCG: * *p* < 0.05 indicated by blue asterisk; HFD + PFDA vs. HFD + PFDA + H-EGCG: *** *p* < 0.001 indicated by red asterisk (panel (**A**)). ** *p* < 0.01 or *** *p* < 0.001 vs. comparison group (panel (**B**)).

**Figure 2 molecules-28-07832-f002:**
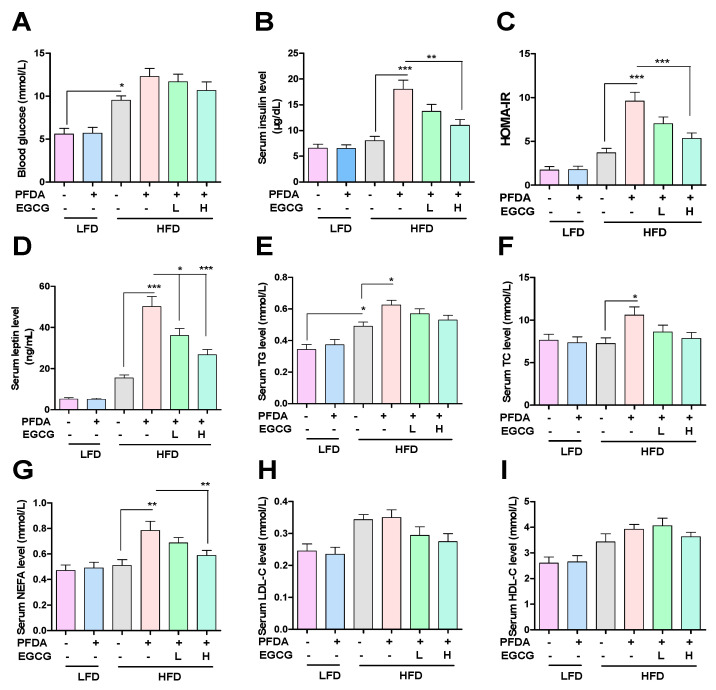
Effects of EGCG on serum parameters in PFDA-exposed mice fed an HFD. (**A**) Blood glucose levels; (**B**) Serum insulin levels; (**C**) HOMA-IR scores; (**D**) Serum leptin levels; (**E**) Serum TG levels; (**F**) Serum TC levels; (**G**) Serum NEFA levels; (**H**) Serum LDL-C levels; (**I**) Serum HDL-C levels. Numbers are the mean ± SEM (*n* = 6/group). * *p* < 0.05, ** *p* < 0.01, or *** *p* < 0.001 vs. comparison group.

**Figure 3 molecules-28-07832-f003:**
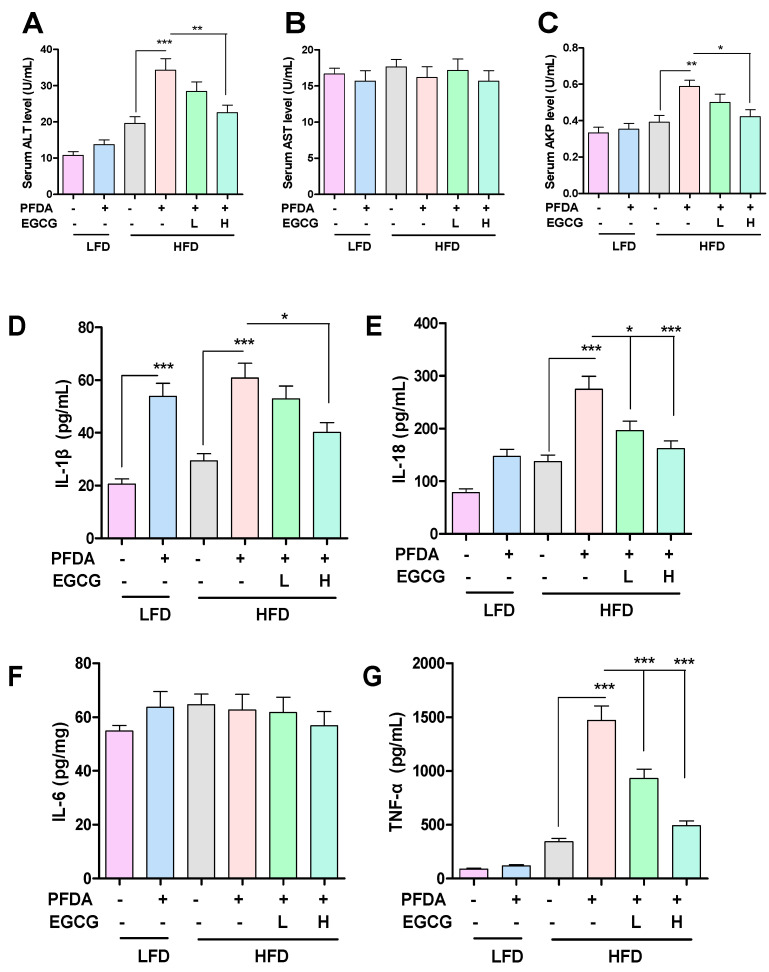
Effects of EGCG on liver function and inflammatory cytokines in PFDA-exposed mice fed an HFD. (**A**) Serum ALT levels; (**B**) Serum AST levels; (**C**) Serum AKP levels; (**D**) Serum IL−1β levels; (**E**) Serum IL−18 levels; (**F**) Serum IL−6 levels; (**G**) Serum TNF−α levels. Numbers are the mean ± SEM (*n* = 6/group). * *p* < 0.05, ** *p* < 0.01, or *** *p* < 0.001 vs. comparison group.

**Figure 4 molecules-28-07832-f004:**
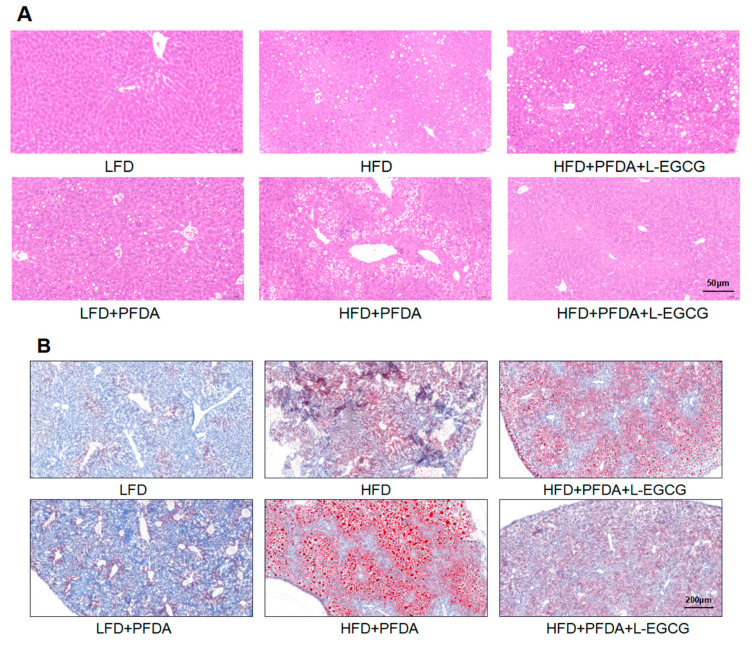
Effects of EGCG on hepatic steatosis and lipid deposition in PFDA-exposed mice fed an HFD. (**A**) Representative pictures of hepatic tissues after H&E staning (20× magnificantion); (**B**) Representative pictures of hepatic tissues after oil red staining (20× magnificantion); (**C**) NAFLD score from histological examination of the liver tissues; (**D**) Liver fat content in terms of percentage, for each diet group. Numbers are the mean ± SEM (*n* = 6/group). For panels (**C**,**D**), ** *p* < 0.01 or *** *p* < 0.001 vs. comparison group.

**Figure 5 molecules-28-07832-f005:**
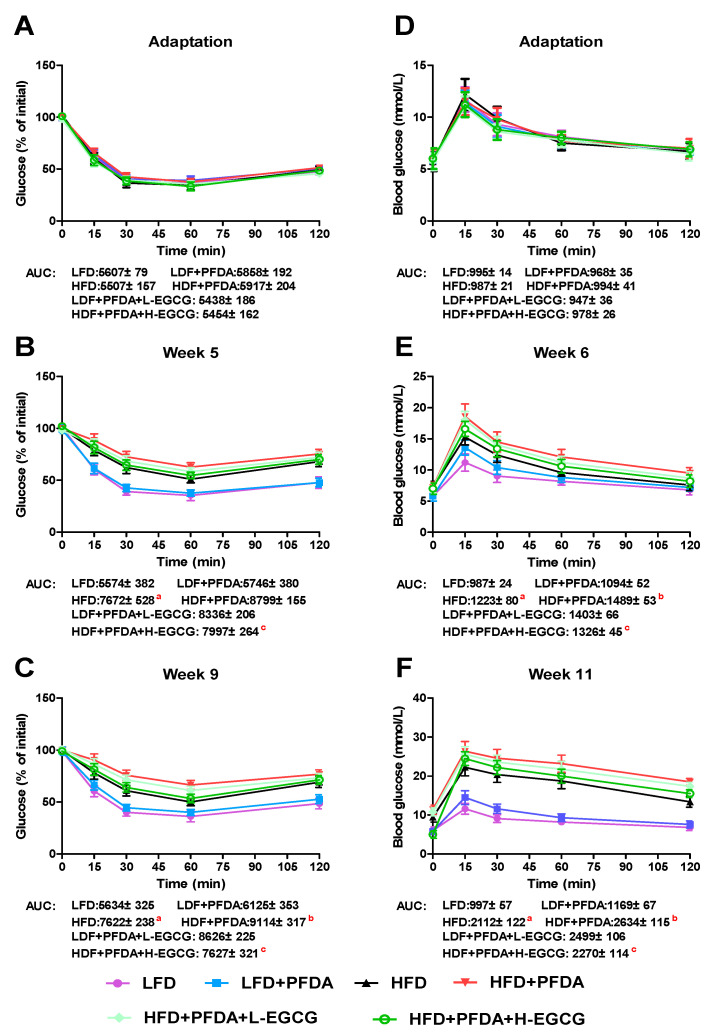
Effects of EGCG on the glucose tolerance test (GTT) and the insulin tolerance test (ITT) in PFDA-exposed mice fed an HFD. (**A**–**C**) GTT; (**D**–**F**) ITT Numbers are the mean ± SEM (*n* = 6/group). For panel (**B**), ^a^ LFD vs. HFD: *p* < 0.05, and ^c^ HFD + PFDA vs. HFD + PFDA + H-EGCG: *p* < 0.05. For panels (**C**,**E**,**F**), ^a^ LFD vs. HFD: *p* < 0.05, ^b^ HFD vs. HFD + PFDA: *p* < 0.05, and ^c^ HFD + PFDA vs. HFD + PFDA + H-EGCG: *p* < 0.05.

**Figure 6 molecules-28-07832-f006:**
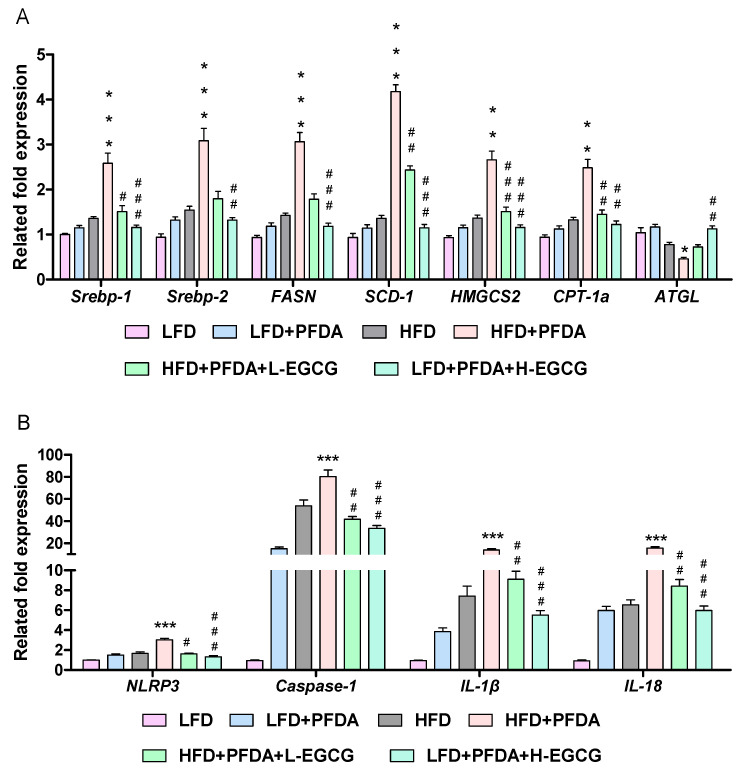
Effects of EGCG on mRNA expression of SREBPs and their target genes and the NLRP3 inflammasome in PFDA-exposed mice fed an HFD. (**A**) mRNA expression of SREBPs and their target genes; (**B**) mRNA expression of NLRP3 inflammasome. Numbers are the mean ± SEM (*n* = 6/group). * *p* < 0.05, ** *p* < 0.01 or *** *p* < 0.001 vs. HFD group; ^#^
*p* < 0.05, ^##^
*p* < 0.01 or ^###^
*p* < 0.001 vs. HFD + PFDA group. *FASN:* fatty acid synthase, *SCD1:* stearoyl-coenzyme A desaturase-1, *HMGCS2:* hydroxymethylglutaryl Coenzyme A synthase 2, *CPT-1a:* carnitine palmitoyltransferase 1A; *ATGL:* adipose triglyceride lipase.

**Figure 7 molecules-28-07832-f007:**
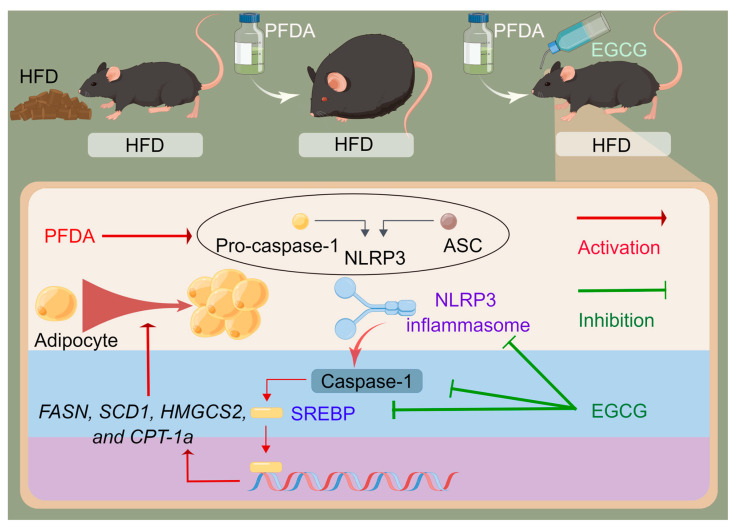
Schematic diagram showing the underlying mechanisms of EGCG in preventing PFDA-exacerbated adiposity and hepatic lipid accumulation in male mice fed an HFD.

**Table 1 molecules-28-07832-t001:** Organ weights (percent of body weight) ^#^.

	LFD	LFD + PFDA	HFD	HFD + PFDA	HFD + PFDA + L-EGCG	HFD + PFDA + H-EGCG
Liver	3.51 ± 0.10	3.46 ± 0.10	3.58 ± 0.16	4.54 ± 0.22 ^b^	4.24 ± 0.18	3.72 ± 0.20 ^d^
Heart	0.44 ± 0.05	0.47 ± 0.06	0.49 ± 0.08	0.44 ± 0.06	0.48 ± 0.10	0.46 ± 0.08
Spleen	0.26 ± 0.04	0.25 ± 0.05	0.29 ± 0.04	0.28 ± 0.03	0.27 ± 0.05	0.28 ± 0.04
Kidney	1.09 ± 0.06	1.12 ± 0.05	1.16 ± 0.06	0.90 ± 0.05 ^b^	0.96 ± 0.06	1.02 ± 0.06
Pancreas	0.47 ± 0.03	0.48 ± 0.03	0.50 ± 0.04	0.42 ± 0.04	0.46 ± 0.05	0.48 ± 0.04
Adopose tissue						
Epididymal	2.10 ± 0.20	2.16 ± 0.30	3.70 ± 0.62 ^a^	5.64 ± 0.48 ^b^	5.15 ± 0.50 ^c^	4.62 ± 0.54 ^d^
Subcutaneous	1.26 ± 0.24	1.32 ± 0.28	2.54 ± 0.40 ^a^	3.78 ± 0.39 ^b^	3.24 ± 0.36	2.64 ± 0.30 ^d^
Retroperitoneal	0.50 ± 0.10	0.52 ± 0.09	0.90 ± 0.12 ^a^	1.82 ± 0.11 ^b^	1.54 ± 0.11	1.26 ± 0.13 ^d^
Mesenteric	1.28 ± 0.14	1.28 ± 0.20	2.12 ± 0.26 ^a^	3.22 ± 0.28 ^b^	2.86 ± 0.24	2.46 ± 0.24 ^d^
Total	5.14 ± 0.45	5.28 ± 0.56	9.26 ± 0.92 ^a^	14.46 ± 1.12 ^b^	12.79 ± 1.04 ^c^	10.98 ± 0.98 ^d^

^#^ Values represent means ± SEM (*n* = 6/group). ^a^
*p* < 0.05 vs. LFD group, ^b^
*p* < 0.05 vs. HFD group, ^c^
*p* < 0.05 vs. HFD + PFDA group, ^d^
*p* < 0.05 vs. HFD + PFDA group.

**Table 2 molecules-28-07832-t002:** Composition of diets.

Ingredient	Low-Fat Diet Amount (g/kg) (LAD 3001M)	High-Fat Diet Amount (g/kg) (TP 02420X)
Casein	139	172
Corn starch	530.992	267.960
Maltodextrin	109	135
Sucrose	82	101
Soybean oil	40	202
Cellulose	50	62
Mineral and vitamin mix	45	55
*L*-Cystine	2	2
Choline bitartrate	2	3
*tert*-Butylhydroquinone	0.008	0.04
Total	1000	1000

**Table 3 molecules-28-07832-t003:** Primer sequences for RT-PCR.

Genes	Direction	Sequences
*SREBP1*	Forward	5′-AGGTGTATTTGCTGGCTTGGT-3′
	Reverse	5′-AGAGATGACTAGGGAACTGTGTGT-3′
*SREBP2*	Forward	5′-AGAAAGAGCGGTGGAGTCCTTG-3′
	Reverse	5′-GAACTGCTGGAGAATGGTGAGG-3′
*FASN*	Forward	5′-GGAGGTTGCTTGGAAGAG-3′
	Reverse	5′- CTGGATGTGATCGAATGCT-3′
*SCD1*	Forward	5′-GCTGGGCAGGAACTAGTGAG-3′
	Reverse	5′-GAAGGCATGGAAGGTTCAAA-3′
*HMGCS2*	Forward	5′-GCCGTGAACTGGGTCGAA-3′
	Reverse	5′-GCATATATAGCAATGTCTCCTGCAA-3′
*CPT-1a*	Forward	5′-GAGACTTCCAACGCATGACA-3′
	Reverse	5′-ATGGGTTGGGGTGATGTAGA-3′
*NLRP3*	Forward	5′-AACAGCCACCTCACTTCCAG-3′
	Reverse	5′-CCAACCACAATCTCCGAATG-3′
*Caspase−1*	Forward	5′-GCACAAGACCTCTGACAGCA-3′
	Reverse	5′-TTGGGCAGTTCTTGGTATTC-3′
*IL−1β*	Forward	5′-CCTGTCCTGCGTGTTGAAAGA-3′
	Reverse	5′-GGGAACTGGGCAGACTCAAA-3′
*IL−18*	Forward	5′-CCTTCCATCCTTCACAGATAGG-3′
	Reverse	5′-CCTGATATCGACCGAACAGC-3′
*GAPDH*	Forward	5′- TCATTGACCTCAACTACATGGT-3′
	Reverse	5′-CTAAGCAGTTGGTGGTGCAG-3′

Note: *SREBP*: sterol-regulatory element binding protein; *FASN*: fatty acid synthase; *SCD1*: stearoyl-coenzyme A desaturase-1; *HMGCS2*: hydroxymethylglutaryl coenzyme A synthase 2; *CPT-1a*: carnitine palmitoyltransferase 1A.

## Data Availability

Data are contained within the article.

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
