# Peer review of "(−)-Epigallocatechin-3-Gallate Reduces Perfluorodecanoic Acid-Exacerbated Adiposity and Hepatic Lipid Accumulation in High-Fat Diet-Fed Male C57BL/6J Mice"

_molecules, 2023, doi:10.3390/molecules28237832_

Round 1

Reviewer 1 Report

Comments and Suggestions for Authors

The study entitled “(-)-Epigallocatechin-3-Gallate Reduces Perfluorodecanoic Acid- Exacerbated Adiposity and Hepatic Lipid Accumulation in High-Fat-Diet Fed Male C57BL/6J Mice” is a very interesting one, including 61 references, however there are some changes to be made:

It turned out to be a very well-researched manuscript and I appreciated that, but I would have liked you to reorganize the Conclusions section. The Conclusions section is far too little detailed. Although the Discussions section is well structured and explained, I would have liked the Conclusions to have been better organized. Please note in Conclusions future perspectives.

Good luck!

Author Response

The study entitled “(-)-Epigallocatechin-3-Gallate Reduces Perfluorodecanoic Acid- Exacerbated Adiposity and Hepatic Lipid Accumulation in High-Fat-Diet Fed Male C57BL/6J Mice” is a very interesting one, including 61 references, however there are some changes to be made: It turned out to be a very well-researched manuscript and I appreciated that, but I would have liked you to reorganize the Conclusions section. The Conclusions section is far too little detailed. Although the Discussions section is well structured and explained, I would have liked the Conclusions to have been better organized. Please note in Conclusions future perspectives.

Response: Thank you for your suggestion. We have reorganized the Conclusions section based on your suggestion.

Reviewer 2 Report

Comments and Suggestions for Authors

The manuscript is well-written and well-presented. My recommendation is to publish it after minor revisions. In particular, I suggest to change from line 47-53 as follows:

As indicated by epidemiological studies [16-18] when exposed to PFDA, humans may be likely to become obese and diabetic. Dietary fat also modulates the metabolism and toxicity of perfluorinated compounds in addition to their direct health hazards effects [15]. There are numerous metabolic diseases associated with a high-fat-diet (HFD), such as obesity, diabetes mellitus, coronary heart disease, fatty liver disease, and stroke [10-12]. These metabolic diseases may be caused by HFD, which contributes to the production of free radicals and the development of inflammation and insulin resistance [13, 14].

I also suggest checking the font and size throughout the manuscript

Author Response

The manuscript is well-written and well-presented. My recommendation is to publish it after minor revisions. In particular, I suggest to change from line 47-53 as follows:

As indicated by epidemiological studies [16-18] when exposed to PFDA, humans may be likely to become obese and diabetic. Dietary fat also modulates the metabolism and toxicity of perfluorinated compounds in addition to their direct health hazards effects [15]. There are numerous metabolic diseases associated with a high-fat-diet (HFD), such as obesity, diabetes mellitus, coronary heart disease, fatty liver disease, and stroke [10-12]. These metabolic diseases may be caused by HFD, which contributes to the production of free radicals and the development of inflammation and insulin resistance [13, 14].

Response: Thank you for your suggestion. We have revised line 47-53 based on your suggestion.

I also suggest checking the font and size throughout the manuscript.

Response: Thank you for your suggestion. We have checked the font and size throughout the manuscript.

Reviewer 3 Report

Comments and Suggestions for Authors

In the study of Xu H. et. al. the authors investigate the effect of different concentration of the major green tea catechin Epigallocatechin -3-gallate (EGCG) on PFDA mediated through high-fat diet obesity. The study is interesting, but the reviewer has some questions that need to be addressed.

-       Title is too long and too descriptive.

-       What is the need of development/testing of treatment that is working only in male C57 mice?

-       The presented experiment is only evaluating the protective effect of EGCG during development of high-fat diet and PFDA mediated obesity and liver damage. The question is will it eliminate obesity. In normal situation a treatment is proposed after appearance of the symptoms and development of the disease conditions. The question here is will EGCG reduce already developed high-fat diet – PFDA obesity.

-       It is not clear the motivation of the authors to choose for the use of EGCG, what about other green tee compounds?

-       The same is related to the applied concentration? Further in respect to the used EGCG concentrations is it mg/kg body weight or diet?

-       In respect to the high-fat diet, why the author chooses to use 20% high fat diet, where the fat is coming from soybean oil? Is it really a high-fat diet? This could be the reason of not significant differences in lipid parameters between LFD and HFD groups. Similar are the observations on the data related to inflammation. A stronger effect is expected to be observed when LFD is compared to HFD.

-       From the whole paper and especially data and discussion it sounds that   

-       Line 280 change “glucose” to “glucose”.

-       It is not really clear the novelty of the mechanism also proposed in Figure 7.

Author Response

In the study of Xu H. et. al. the authors investigate the effect of different concentration of the major green tea catechin Epigallocatechin -3-gallate (EGCG) on PFDA mediated through high-fat diet obesity. The study is interesting, but the reviewer has some questions that need to be addressed.

-Title is too long and too descriptive.

Response: revised.

-What is the need of development/testing of treatment that is working only in male C57 mice?

Response: PFDA can affect the fat metabolism of female animals by interfering with estrogen metabolism, thereby increasing the complexity of research.

-The presented experiment is only evaluating the protective effect of EGCG during development of high-fat diet and PFDA mediated obesity and liver damage. The question is will it eliminate obesity. In normal situation a treatment is proposed after appearance of the symptoms and development of the disease conditions. The question here is will EGCG reduce already developed high-fat diet – PFDA obesity.

Response: Thank you very much for your question. Unfortunately, we have not conducted relevant studies, accordingly, we have added a paragraph to indicate this question as a study limitation.

-It is not clear the motivation of the authors to choose for the use of EGCG, what about other green tee compounds?

Response: Thank youfor your comments. We choose EGCG for two reasons. The first reason is that EGCG is the most abundant catechin of green tea, accounting for 50-70% of 50%–80% of all catechins. The second reason is that our previous research found that EGCG has a protective effect on high-dose PFDA induced liver injury in mice (Food Res Int. 2020:127:108628. doi: 10.1016/j.foodres.2019).

-The same is related to the applied concentration? Further in respect to the used EGCG concentrations is it mg/kg body weight or diet?

Response: Thank youfor your comments. It is mg/kg body weight of each mouse.

-In respect to the high-fat diet, why the author chooses to use 20% high fat diet, where the fat is coming from soybean oil? Is it really a high-fat diet? This could be the reason of not significant differences in lipid parameters between LFD and HFD groups. Similar are the observations on the data related to inflammation. A stronger effect is expected to be observed when LFD is compared to HFD.

Response: Thank you for your comments. The rationale behind selecting a 20% fat content is to facilitate a more comprehensive examination of the alterations ensuing from the combination of PFDA and a high fat diet. Employing a 40% fat diet regimen would not yield a substantial exacerbating impact when combined with PFDA. The selection of a 20% fat diet as a means to investigate the impact of exogenous toxins on adiposity and hepatic lipid accumulation in mice is a well-established and widely employed experimental model, as evidenced by its utilization in our own (Food Chem Toxicol. 2023;178:113943. doi: 10.1016/j.fct.2023.113943.) and other researcher (J Agric Food Chem. 2016 ;64(49):9293-9306. doi: 10.1021/acs.jafc.6b04322.) publications. Choosing soybean oil also referred to these above published literature.

-From the whole paper and especially data and discussion it sounds that

Line 280 change “glucose” to “glucose”.

Response: revised.

-It is not really clear the novelty of the mechanism also proposed in Figure 7.

Response: We have revised Figure 7 to clear novelty of the mechanism.

Round 2

Reviewer 3 Report

Comments and Suggestions for Authors

The authors addressed well the reviewer comments.

Still Figure 7 need more clarifications. It is good that the authors choose to represent the effect of EGCG with green and the one of PFDA with red. Please use those colors to make the proposed mechanism of actions with the respective compounds/colors more clear on the different levels.

Author Response

Response: 

Dear Reviewer, thank you for your rapid response on our manuscript. The constructive criticism of you was much appreciated and we revised Figure 7 accordingly in the revised manuscript. 
